# Abstract: Enhancing Access Control in EUDI Wallets with Solid Principles to Prevent Fraud

Authors:
Jan Lindquist (Swedish Institute for Standards - Privacy Standards Developer)
Harshvardhan J. Pandit (Dublin City University)

## Objective

The European Digital Identity (EUDI) Wallet initiative aims to provide secure, privacy-preserving digital identity management. However, ensuring robust access control mechanisms to prevent fraud remains a significant challenge. This paper explores how integrating principles from the Solid Project can enhance access control within EUDI Wallets, empowering users while maintaining compliance with GDPR and eIDAS regulations.

## Existing Landscape and Challenges

EUDI Wallets are designed to store and manage personal identity data, qualified and non-qualified attestations, and transaction logs. Currently, access control in these wallets relies on predefined policies and user consent mechanisms. However, there are gaps in fraud prevention, particularly regarding relying party (RP) verification, incomplete disclosure of critical metadata, and the absence of role-based enforcement. The lack of transparency in RP requests and insufficient validation of request parameters leave users vulnerable to identity misuse and fraud.

## Addressing Gaps: The Role of Solid Principles

Solid (Social Linked Data) promotes decentralized data control, allowing users to manage their personal data through secure and interoperable personal online data stores (Pods). Applying Solid principles to EUDI Wallets can enhance fraud prevention by introducing:

- **Granular Access Control**: Users define access permissions dynamically, limiting data exposure to legitimate RPs.
- **Linked Data Transparency**: RPs must include verifiable claims and metadata, ensuring accountability in identity verification processes.
- **Decentralized Verification**: Cross-checking RP credentials against trusted lists (e.g., EU/EEA Trusted List Browser) before granting access.

# Proposed Approach: Wallet Access Control Engine (WACE)

 To operationalize these principles, we introduce the Wallet Access Control Engine (WACE), a functional abstraction that evaluates RP requests based on both issuer verification and adherence to predefined codes of conduct. WACE integrates real-time fraud prevention checks by:

- **Validating RP authenticity**: Ensuring the inclusion of recipient details, purpose hints, and RP numbers to mitigate impersonation risks.
- **Implementing role-based access control (RBAC)**: Cross-referencing role types with EU-recognized trust lists.
- **Generating user-centric notifications**: Informing users about potential risks, such as data misuse or insufficient RP transparency.

The following diagram is part of a CEN TC224 WG20 Digital Wallet project called "EUDI Wallet Held Assets Access Control" that introduces WACE.

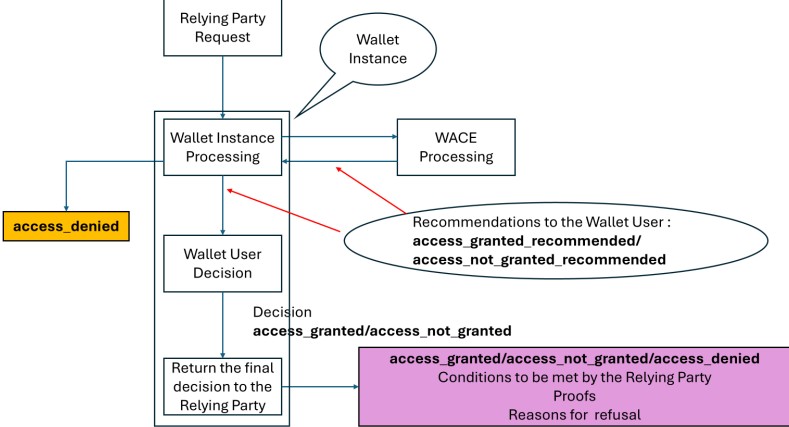

# Findings and Practical Considerations

Initial evaluations suggest that WACE can significantly reduce fraud risks by enhancing user awareness and enforcing stricter validation checks on RP requests. Practical considerations include:

- Adoption of Solid-based data governance within EUDI Wallet architecture.
- Regulatory alignment to ensure compliance with GDPR and eIDAS.
- Collaboration between wallet providers, RPs, and regulators to establish interoperable fraud prevention mechanisms.

# Conclusion

By integrating Solid's user-controlled data management approach with EUDI Wallet access control mechanisms, we create a more secure and transparent identity verification framework. This approach minimizes fraud risks while maintaining interoperability, user sovereignty, and regulatory compliance. Future research will focus on extending WACE's capabilities to detect emerging fraud patterns and standardizing access control policies across digital identity ecosystems.

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
