# OpenReview forum: "Enhancing Access Control in EUDI Wallets with Solid Principles to Prevent Fraud"
_SolidProject.org/SoSy/2025/Privacy_Session — SoSy2025-Privacy_

### Official Review · ~Jesse_Wright1 · 2025-03-22
**WACE may be a promising contribution to EUDI space - however paper lacks sufficient description of WACE architecture, and the relationship to Solid is loose.**

**Rating:** 5
**Confidence:** 4

**Review:**

Whilst I agree with the premise that Solid is applicable as a holder service in digital wallet spaces such as EUDI (c.f. https://blog.jeswr.org/2025/02/14/data-wallets), it is unclear from this paper how WACE relates to Solid, or what access control mechanisms it uses.

I would weakly accept on if the authors can:
 - Fully present the WACE architecture during the symposium, and ideally make the codebase public
 - Be transparent about whether there is any relationship to Solid (I would still moving my rating from 5->6 in the ‘no’ case provided the above point is satisfied)

For specific feedback:

The section “Addressing Gaps: The Role of Solid Principles” is not compelling in particular on the points:

> Granular Access Controls

Whilst Solid uses WAC/ACP controls to define resource access - in most cases credentials are requested using a verifiable presentation query endpoint. It is unclear whether WAC/ACP - or even for a stretched relationship to Solid, ODRL - are operationalised in your framework.


> Linked Data Transperancy

EUDI is build upon verifiable credential standards, so the verifiable claims and metadata is a given.

> Decentralised Verification

The link to Solid is not immediately clear. It also appears that this is just implementing the EUDI requirement to use Trusted Service Providers.

> Proposed approach

The following are unclear from the papers description
RP Aiuthenticityy
RBAC - is this not just implementing an EUDI regulatory requirement
How are the data misuse risks identified (at a guess, excessive ODRL permissions requested?)


> Practical considerations

These are very high level. Specific questions include:
 - What do you mean by Solid-based data governance, and how do you plan to include this in the EUDI architecture
 - What regulatory alignment are you looking for? As an example, in the UK https://blog.jeswr.org/2025/02/14/wallet-governance has been suggested for the similar digital verification scheme.
 - What kind of collaboration is needed for fraud prevention. I assume you mean identity fraud protection from stealing credentials. The regulation around trust services already does a lot of work for this in terms of certifying services, and having minimum requirements for identity checks - what is missing?

> Other notes


Please provide a clearer system description, and access to the codebase if possible.

As the authors discuss access grants, which we guess may possibly be ODRL encoded, we wish to alert the author to the fact that W3C Verifiable Credential standards now include a termsOfUse field (https://www.w3.org/TR/vc-data-model-2.0/#terms-of-use)

---

### Official Review · ~Ines_Akaichi1 · 2025-03-25
**WACE seems an appealing solution to EUDI wallets but the approach raises several questions.**

**Rating:** 5
**Confidence:** 4

**Review:**

This abstract presents a proposal to enhance access control for EUDI wallets using SOLID principles. Enhancing access control in EUDI wallets seems an appealing idea. However, the connection to SOLID is not fully defined; in particular the various principles that are meant to improve access control in SOLID.

Below, I cite several concerns and questions regarding the proposed approach:

1.	The authors suggest an access control mechanism based on policies managed by users rather than predefined policies. However, how would users define these policies without knowledge of or access to potential relying parties? This seems to contradict the motivation for introducing WACE in the first place. Wouldn't users be better served with predefined policies instead?
2.	The proposed approach appears to rely on role-based access control. However, in a decentralized environment, role-based models are typically suited for closed systems with well-defined roles. Attribute-based access control (ABAC) might be a better fit, as it allows for more flexible and metadata-driven access decisions, especially considering that the relying party’s identity can be described using various attributes. Why were role-based access chosen over ABAC in this context?
3.	The abstract mentions risk notifications, but it is unclear what kind of information would be shared with users. Simply informing users of risks without providing guidance on how to mitigate them may not be very helpful. Have the authors considered how users can be supported in managing these risks?
4.	I am missing an architecture where the interplay between SOLID and EUDI wallets are shown.
5.	The accompanying figure is unclear. It could be improved by explicitly depicting the different entities involved and using arrows to illustrate the flow of information, such as the request from the relying party, the access decision, and other key interactions.

Overall, while this is just an abstract, it raises several questions (see above) that I believe should be addressed if the paper will be accepted.

---

### Decision · Program_Chairs · 2025-04-01

Accept